# Optimization of technique parameters of pneumatic molding for rice straw bowl tray

Hailiang Li[1,2☯], Chun Wang[1,3], Haitian Sun[1,3], Huafen Zou[1], Hongxuan Wang[1], Huiyuan Wang[4], Xinyu Liu[1,5☯], Zhenzhen Yu[1,3]*

1 Hainan Provincial Key Laboratory of Tropical Crop Nutrition, South Subtropical Crop Research Institute Chinese Academy of Tropical Agricultural Science, Zhanjiang, China, 2 College of Mechanical and Electrical Engineering, Henan Agricultural University, Zhengzhou, China, 3 College of Engineering, Heilongjiang Bayi Agricultural University, Daqing, China, 4 Institute of Scientific and Technical Information, Chinese Academy of Tropical Agricultural Sciences, Haikou, China, 5 Renhe Meteorological Bureau of Panzhihua, Panzhihua, China

☯ These authors contributed equally to this work.
* yudq1994@hotmail.com

**Data Availability Statement:** All relevant data are within the paper and its Supporting information files.

**Funding:** Hainan Provincial Postdoctoral Science Foundation (322QN375); Special Fund for Basic

## Abstract

In order to find out the optimized technological parameters of rice straw bowl tray, the forming test was carried out with the slurry made of rice straw as the main raw material and the pneumatic molding machine as the equipment. The orthogonal rotational combination test of ternary quadratic regression and response surface analysis method were used to study the effect of 3 molding factors (vacuum degree, pressure preservation time and adsorption time) on 3 molding technical indices (bowl hole molding rate, relaxation density and rupture-resisting strength). Design-Expert data analysis software was used to establish regression models between rice straw bowl tray molding factors and molding properties so as to obtain optimal technological parameters, which were as follows: vacuum degree: -0.09 MPa; holding time: 14 s; adsorption time: 5 s. At the moment, theoretical bowl hole molding rate was 98.62%, relaxation density was 5.8 g/cm3 and rupture-resisting strength was 27.48 N, the experimental results show that the theoretical analysis is correct, and the model fitting is better. This study can provide a theoretical basis for optimization of pneumatic molding technological of rice straw bowl tray and lay a foundation for realizing industrialized production of biomass seeding tray.

## 1 Introduction

Seedling transplanting is an effective measure to improve rice yield in cold regions of China, and rice seeding bowl tray is a essential equipment in the process of seedling transplanting [1, 2]. The commonly used seeding bowl tray is made of polyethylene, which have some advantages such as low price, light weight and good water resistance. However, it will damage the root system during transplanting, which is not easy to degrade, and improper recycling will cause environmental pollution [3, 4]. Biomass seedling tray is mostly made from agricultural waste resources such as crop straw, livestock and poultry manure and other raw materials [5–

Scientific Research Expenses of the Chinese Academy of Tropical Agricultural Sciences (19CXTD-31).

**Competing interests:** The authors declared no potential conflicts of interest with respect to the research, author- ship, and/or publication of this article.

8], which is not only improves the utilization rate of agricultural waste, but also solves the problem of environmental pollution. Biomass seedling tray has the advantages of not hurting seeding roots, biodegradable and self-raised, effectively improving the soil physical and chemical nature, and improving the soil organic matter content [9–13], it is gradually accepted by farmers in the process of promotion and application, and the research of biomass seedling tray is gradually becoming a hot topic of research.

Enhuui Sun et al., took rice husk and corn starch as the main raw materials and prepared biomass seeding trays by urea aldehyde modification, the thermal degradation behavior of different seedling containers was quantified by thermal weightlessness, the results showed that the compound urea formaldehyde-maize starch adhesive is 108.9% stronger than the maize starch adhesive and has good biodegradation properties [14]. Biomass seeding trays were made using the sunflower seed and rice shell as the substrate by P. Postemsky et al., and evaluated the role of two types of seedling bowl tray in tomato seedling transplanting, cultivate seedlings and production, which showed that the effect of sunflower shell seeding bowl tray on tomato growth and development, flowering and early fruit production were comparable to the control group [15]. Ping Qu prepared a degradable seedling bowl tray by mixing the hydrolyzed soybean separation protein modified urealdehyde resin with straw powder and analyzed the tensile strength and degradability of the biomass seeding tray. The results showed that modified urealdehyde resin can provide nitrogen source for crop growth during degradation and is more beneficial to plant growth [16]. Guofeng Wu studied a biodegradable seedling bowl tray made of straw, starch and polyvinyl alcohol modified starch as adhesive, and characterized the biodegradation properties by hygroabsorption, infrared spectroscopy, degradability, and thermal weightlessness analysis, which showed that polyamide resin plays a positive role in promoting the degradation of seedling bowl tray [17]. Li Lianhao changed the mold's translational motion to circular motion, designed a new type of continuous production system for seeding-growing bowl tray. The optimal process parameters were obtained by using the weight optimization analysis method: the rotariona line speed of the forming roll was 2.5 m·min$^{-1}$, straw content reached 65%, the thickness of the mixing was 5mm, and the distance of retreating wire rod position was 4mm [18]. Wan Pengju prepared a kind of biomass seedling bowl for eggplant and fruit vegetables with fermented animal feces and rice straw as molding materials. It showed that the molding quality of biomass seedling bowl was the best when the molding pressure was 126.1KN, the molding temperature was 141.1˚C, the moisture content of material was 12%, and the mass fraction of straw was 6% [19]. Liu Dejun took the moisture content, compression ratio and compression speed as the test factors, carried out the ternary quadratic orthogonal rotation combination test design, established the mathematical model between the moisture content, compression ratio, compression speed and water absorption, anti destruction strength, expansion rate and other parameters, and obtained the optimized parameter combination of each index as follows: the moisture content is 22%, the compression ratio is 2.9, and the compression speed is 90mm/min. At this time, the seedling block has strong anti destruction strength and good water absorption performance, low expansion rate [20]. Through experimental research and analysis, Ma Yongcai obtained the influence of key factors on the performance evaluation indexes of the bowl tray during the compression molding process, and obtained the optimized parameter combination of compression molding as follows: the molding pressure is 26 MPa, the mass ratio of straw is 16.44%, the mass ratio of solid coagulant is 6.26%, the mass ratio of water is 19.96%, and the biological starch glue is 0.04% [21]. Xiao Yuan used citrus reticulata blanco as raw material and polylactic acid as adhesive, the degradable nursery bowl of citrus peel residue was prepared by molding method. The preparation process parameters of citrus peel residue / polylactic acid degradable nursery bowl were optimized by response surface methodology, the optimized process scheme

was as follows: citrus peel residue particle size 20 mesh, upper mold temperature 118˚C, lower mold temperature 133˚C, molding pressure 2 MPa, and molding time 47 s [22]. Wang Chun used the crushed crop straw, thermofixation adhesive and curing agent as raw materials, using thermal pressure forming technology prepared rice straw bowl tray. The relevant tests showed that the bowl tray has the advantages of increasing production, protecting the environment and promoting the modernization of rice production [23–28].

It can be seen from the development status at home and abroad that there are various kinds of biomass seedling bowl, but the molding technology and molding process are all studied for the compression molding, while the research on pneumatic molding process has not been reported [17, 29, 30]. Pneumatic forming is a method of forming products by using positive pressure or negative pressure formed by pneumatic force, the role of gas replaces the forming part of some molds, so the large-size forming parts can be obtained by using simple forming equipment, without heating or pressurization, with lower energy consumption, and can be used for the preparation of complex shaped parts [31]. The research team creatively applied the pneumatic forming technology to the preparation process of the rice straw bowl tray, and developed the rice straw bowl tray forming machine [32], but the relevant forming process has not been studied.

Based on the above problems, a rice straw bowl tray production test with vacuum degree, pressure preservation time and adsorption time as the test factors, bowl hole molding rate, relaxation density and rupture-resisting strength as the evaluation index was conducted using existing equipment, analyzed the obtained data using Design Expert analysis software, explored the influence of test factors on the test index, and provided theoretical basis and reference for obtaining the optimal process parameters of rice straw bowl tray pneumatic forming.

## 2 Materials and method

### 2.1 Test materials

The biomass raw materials used in the test were rice straw and air-dried cow dung collected at the south subtropical crop research institute of the Chinese Academy of Tropical Agricultural Science in October 2020, which was crushed into 5–10 nets. Injected the crushed straw and cow dung into the pulping tank at a mass ratio of 2:1, and the slurry at a concentration of 30% was made by adding water, and added 530g biomass adhesive (According to the pulping tank volume of 1.5 m$^3$).

### 2.2 Testing equipment

TL-YPCX-01 Pneumatic Molding Machine for rice straw bowl tray (hereinafter abbreviated pneumatic molding machine) was used for bowl tray molding, and the equipment mainly consisted of molding system, drive system, pulping system and control system, overall structure is shown in Figs 1 and 2, main technical parameters are shown in Table 1.

The pulping system mainly included seriflux stir pot, mixing tank, ventilation pipe, etc. The molding system mainly includes vacuum pump, air compressor, cylinders, and molds, etc. The main and deputy molds are composed of air flow distribution room, pneumatic panel, template, seal strip and ventilation pipe, etc. (Fig 3). The convex formwork of the main mold is a mesh structure, which can ensure the passage of water and air and leave the solid material in the slurry on the surface of the convex formwork, the airflow exchange between the airflow distribution room and the outdoor air can be realized through the open hole. The main mold is fixed in the slurry storage tank through the mold bracket, relying on the vertical cylinder in the vertical direction to realize the conversion of mold immersed into the slurry and removal from the slurry, so as to form and shape the rice straw bowl tray; The deputy mold relies on

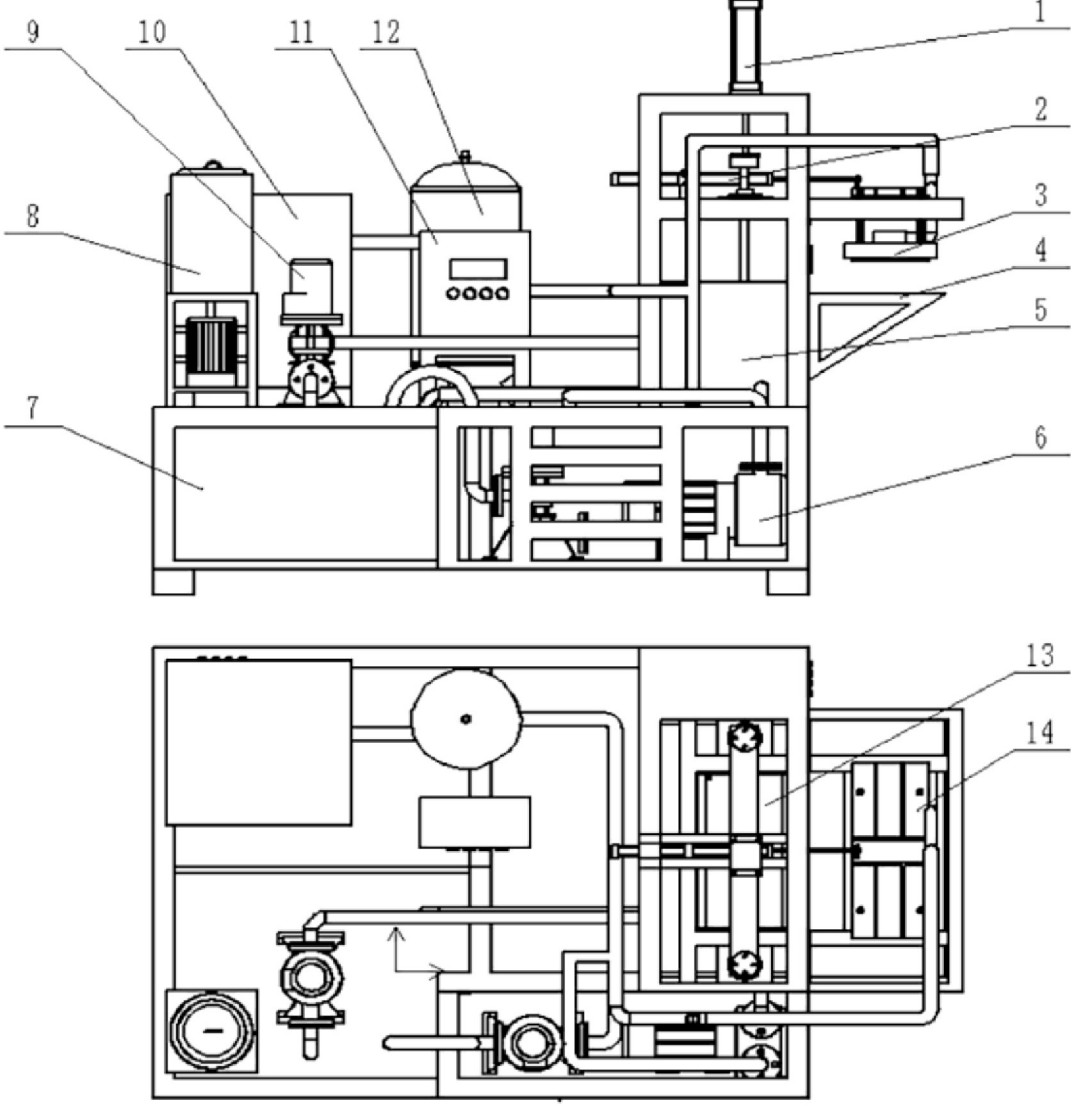

**Fig 1. Structural schematic diagram of the pneumatic molding machine.** 1. Vertical cylinder 2. Horizontal cylinder 3. Deputy mold 4. Objective table 5. Slurry storage tank 6. Water-ring vacuum pump 7. Pulping tank 8. Mixing tank 9. Water pump 10. Air compressor 11. Control cabinet 12. Gasholder 13. Main mold 14. Mold bracket.

the horizontal cylinder to complete the horizontal movement, which can cooperate with the main mold to achieve rice straw bowl tray shape and demoulding. The initial position of the deputy mold is directly above the main mold and the templates of the main mold and the deputy mold are engaged with each other. Vacuum pump and air compressor are connected to the mold through the ventilation pipe to realize the conversion of positive and negative pressure through the solenoid valve control, and finally characterized as the blow and suction on the template surface.

Type MFKP666X45 pulverizer (Jinan Yuezhen Machinery Co., Ltd.) is used to crush straw, WDW-200E microcomputer control electronic universal test machine (Jinan Time Gold Test Machine Co., Ltd.) is used for the stress test on the rice straw bowl tray, Type YHMW900-100 microwave hot wind coupled multi-function dryer (Heilongjiang Bayi Agricultural University)

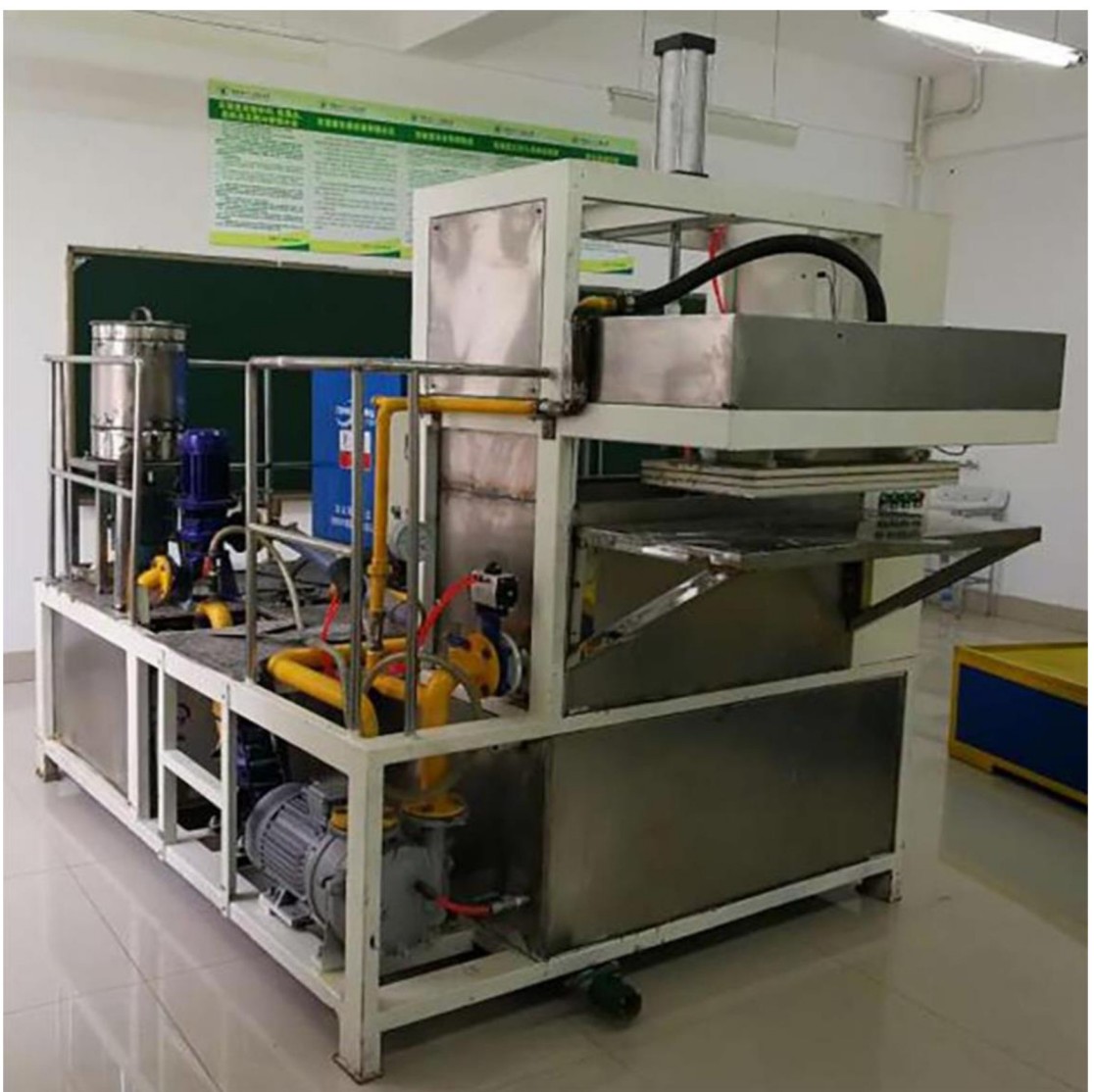

**Fig 2. Physical diagram of pneumatic molding machine.**

**Table 1. Structure parameters of pneumatic molding machine.**

| Measurement | Parameter |
|---|---|
| Length×width×height/mm×mm×mm | 2900×1800×2380 |
| Full load power/kW | 20 |
| Vacuum range/MPa | 0~-0.1 |
| Pulping tank volume/$m^3$ | 1.5 |
| Work efficiency/tray·$day^{-1}$ | 1200 |

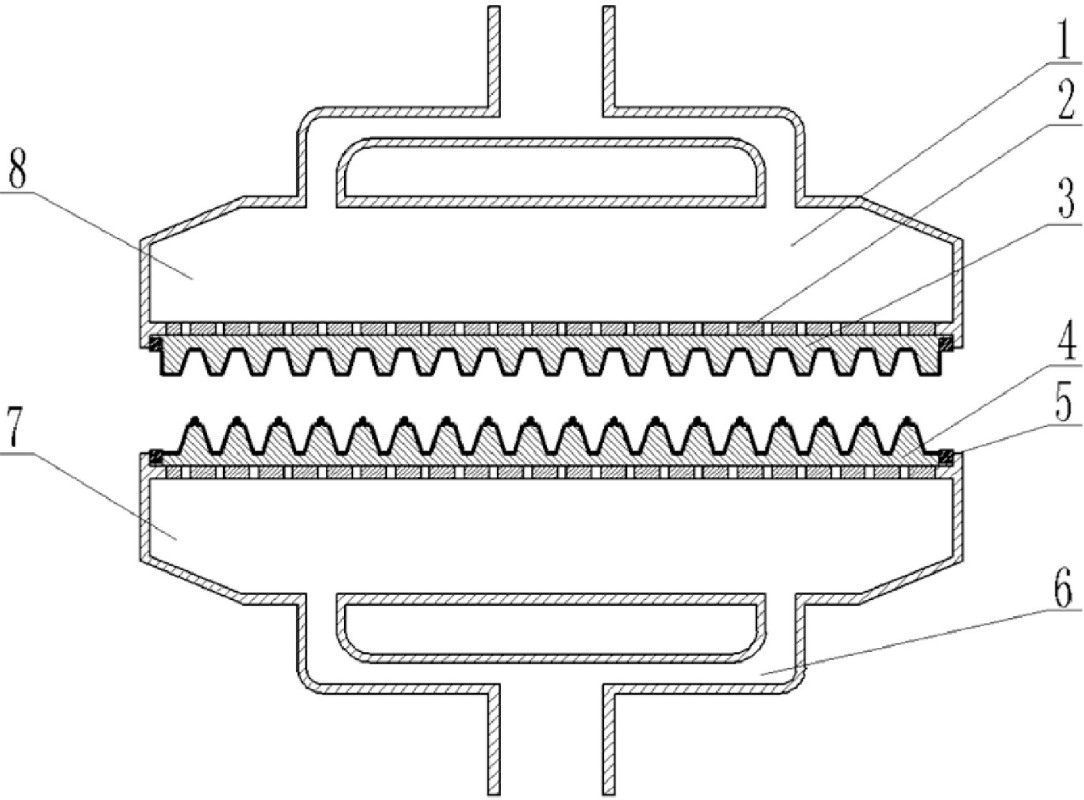

**Fig 3. Brief diagram of molding mold structure.** 1. Airflow distribution chamber 2. Pneumatic panel 3. Concave template 4. Convex template 5. Seal strip 6. Ventilation pipe 7. Main mold 8. Deputy mold.

is used for drying rice straw bowl tray, and other equipment also included BSM5203 electronic scale (range: 0~520g, precision 0.001g), cursor caliper, stopwatch, cutting tool, etc.

## 2.3 Test method

**2.3.1 Production method of rice straw bowl tray.** The work flow of the pneumatic molding machine is shown in Fig 4. The pretreatment straw and cow dung were put into the

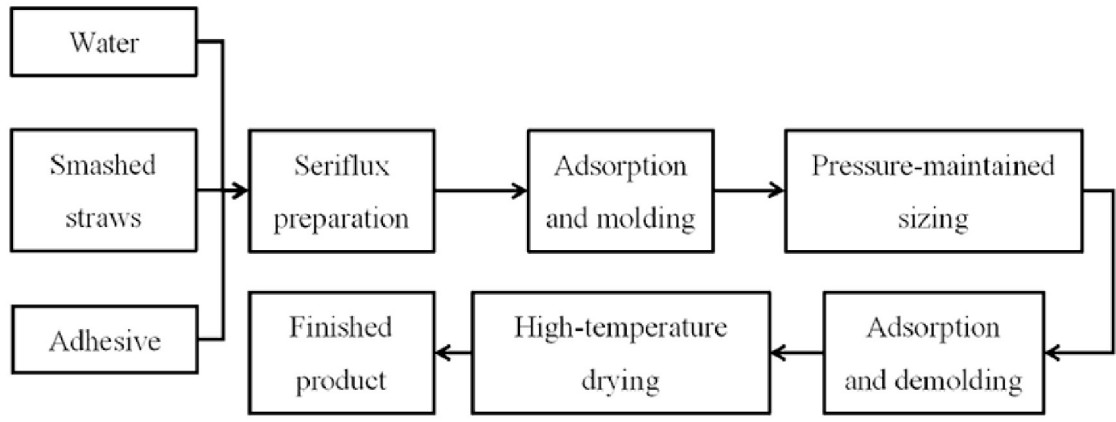

**Fig 4. Preparation processing of rice straw bowl tray.**

pulping tank according to the mass ratio of 2:1, adding water to make a 30% concentration of slurry, and mixed evenly with 530g of adhesive to obtain the raw materials for the preparation, the mass ratio of straw to the whole material is called the straw quality score. Start the pneumatic molding machine, the vacuum pump produces negative pressure in the main mold cavity, the adsorption force adsorbs the solid material in the slurry to the surface of the mesh template. The main mold is raised upward to the liquid surface under the action of the vertical cylinder. At this time, the material on the template surface is pressure preserve by the adsorption force, and the main mold continue to be raised to engage with the deputy mold after a certain period of time, the adsorbent material was further molded by the squeezing of the two molds. At the same time, the main mold connected to the air compressor produces air blowing force, the deputy mold connected to the vacuum pump produces adsorption force, and the blank is separated from the main mold under the resultant force and adsorbed on the surface of the deputy mold. Afterwards, the main mold returns to the initial position for the start of the next working cycle, the deputy mold moves horizontally directly above the objective table under the action of the horizontal cylinder. This moment, the air compressor connected to the deputy mold is started, the template surface produces air blowing force, the blank fall to the objective table under the blowing force and its own gravity, which completes the shaping cycle of a rice straw bowl tray. Vacuum pump and air compressor are connected to the mold through the ventilation pipe, and the connection relationship conversion is realized through the control of the solenoid valve to realize the positive and negative pressure conversion in the airflow distribution chamber. The whole preparation process can be automated under the action of the intelligent control system, without manual operation.

**2.3.2 Experimental factors.** Vacuum refers to the degree of gas sparsity in the airflow distribution chamber. The model forming mainly depends on the adsorption force generated by the negative pressure, the level of vacuum in the airflow distribution chamber directly determines the strength of the adsorption force generated on the template surface. When the adsorption force is too large, the solid material adsorbed on the template surface is increasing, and the thickness of the side wall and bottom surface of the rice straw bowl tray is increasing, which is not easy for the development of seedling roots. However, the adsorption force is too small to absorb enough material, the thickness of the rice straw bowl tray is reduced, the straw bowl damage resistance strength is reduced, and the bowl hole molding rate is reduced. Therefore, determining a reasonable vacuum degree is the key to rice straw bowl tray forming. According to the preliminary test, the mold vacuum degree is suitable within the range of-0.05~-0.1GPa.

Adsorption time is the length of time when the mold adsorbs the materials in the slurry storage tank. The longer the adsorption time, the tighter the rice straw bowl tray, the higher the damage resistance strength, but the greater the required power consumption. The shorter the adsorption time, the smaller the rice straw bowl tray tightness, and the lower the bowl hole molding rate. The preliminary test results showed that the adsorption time is the most suitable in the range of 2 to 6s.

The pressure preservation is carried out after the material suction mold is completed and the mold is removed out of the slurry storage tank, the pressure preservation time and vacuum degree are the main factors that determine the shaping effect. In the process of pressure preservation, the moisture content of the rice straw bowl tray gradually decreased, and the density gradually increased. The longer the pressure preservation time, the higher the relaxation density of the rice straw bowl tray, the greater the adhesion force between the rice straw bowl tray and the template, the more difficult the rice straw bowl tray to demold. The pressure preservation time is short, the rice straw bowl tray relaxation density is small, the low rupture-resisting strength, easy to cause the damage of the rice straw bowl tray, affecting the bowl hole molding

rate. The results of the preliminary test showed that the appropriate time range of pressure preservation is 10 to 20s.

**2.3.3 Test indicators and measurement methods.** The forming rate and strength are important indicators to evaluate the production efficiency and use performance of rice straw bowl tray, since the evaluation criteria of biomass seedling tray have not been put forward, the bowl hole molding rate, relaxation density and rupture-resisting strength are selected as the indexes for evaluating the forming effect of the rice straw bowl tray by consulting the relevant information and combined with the actual rice production situation [33–36].

The bowl hole molding rate is used to measure the forming effect of rice straw bowl tray, the higher the bowl hole molding rate, the better the forming effect and the higher productivity. The preliminary test results showed that when the bowl hole molding rate is more than 98%, it can meet the production requirements of the rice straw bowl tray. During the preparation of rice straw bowl tray, the forming effect is affected by various factors such as the process and raw material ratio, in order to facilitate statistical analysis, the bowl with depth of more than 1/2 of the theoretical depth is defined as the qualified bowl [28], and after averaging 30 cumulative trials, they are calculated using Eq (1).

$$K = 1 - \frac{K_1}{18360} \times 100\% \tag{1}$$

where, $K$ is bowl hole molding rate, %; $K_1$ is number of unqualified bowls in 30 rice straw bowl tray, a; 18360 is total number of all bowls in the 30 rice straw bowl tray, a.

Relaxation density refers to the density when the molding will gradually decrease after unloading and remolding due to elastic deformation and stress relaxation, and tends to stabilize after a certain time [37, 38]. The higher the relaxation density, the better the molding stability, the smaller the probability of disintegration rice of straw bowl tray during seeding raising, but its porosity and permeability become worse, which inhibit the transport of water, fertilizer, gas and heat, and is not conducive to the growth and development of seedlings [12]. After preliminary tests, the rice straw bowl tray relaxation density is at 4~5.8g/cm³ can meet the requirements of seedling raising and strength requirements. In this test, the formed and dried rice straw bowl trays were immersed in water for 48h to measure the mass and volume changes of the rice straw bowl trays before and after water immersion, and form the formula (2) by the derivation and transformation of the buoyancy and density calculation formula.

$$\rho = \frac{m_1}{\Delta V + m_2 - m_1} \tag{2}$$

Where, $\rho$ is relaxation density, kg/m³; $m_1$ is rice straw bowl tray quality before immersion, kg; $m_2$ is rice straw bowl tray quality after immersion, $kg$; $\Delta V$ is the volume change of water after immersion, m³.

The rupture-resisting strength is used to measure the physical quality characteristics of the rice straw bowl tray, mainly divided into hard strength and soft strength. Hard strength determines the damage rate in the transportation of rice straw bowl tray, the greater the strength, the lower the damage rate. The strength requirements can be meet when the fracture stress reaches 20N. Soft strength determines the transplanting effect of the rice straw bowl tray, the smaller the strength, the less the kinetic energy needed by the transplanting needle, the better the transplanting effect. Through the literature [39], the strength of seedling needle extraction is generally between 13.4N~21.2N, the results of the preliminary test showed that the fracture stress range of soft rice straw bowl tray is 0.7N~1.2N, which is far less than the required stress value, the soft strength of the rice straw bowl tray can meet the requirements of transplanting, therefore, the hard strength is the assessment index in the test design process.

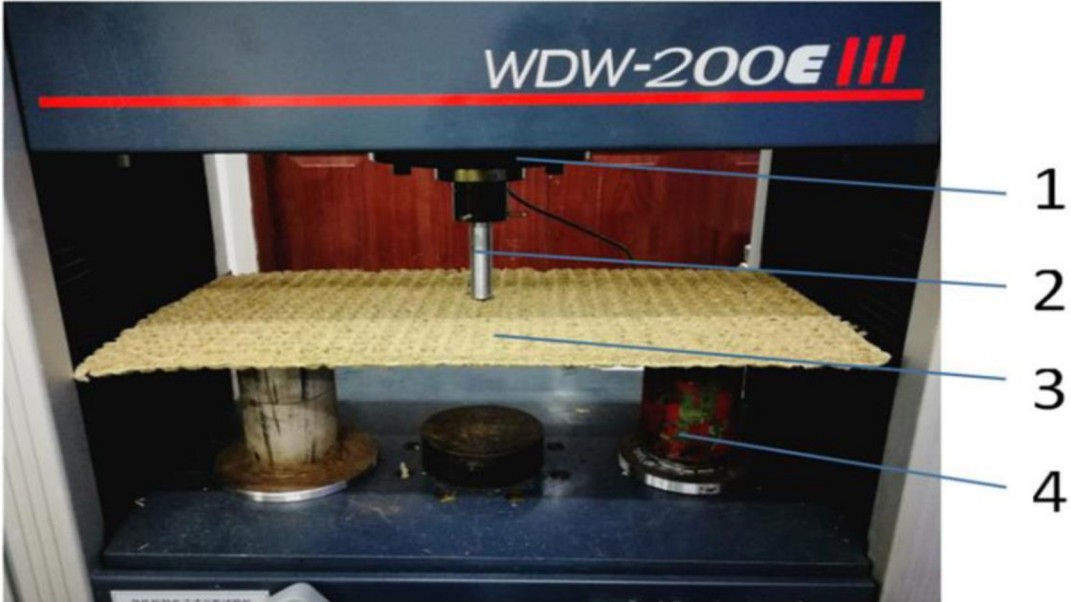

**Fig 5. Test of anti destructive strength of rice straw bowl tray.** 1. Platen 2. Pressure rod 3. Rice straw bowl tray 4. Support frame.

The maximum stress of the rice straw bowl tray before the fracture is an important parameter of its mechanical strength, corresponding to the destructive of the macroscopic structure [40]. In this study, the rupture-resisting strength indicates by the critical pressure of the rice straw bowl tray. The rice straw bowl tray is placed on the support frame of the loading table (face up and middle is suspended), right above the center of the rice straw bowl tray is a cylindrical pressure rod, the destructive strength test is conducted with the universal test machine, the pressure load is increased at a fixed rate of 10mm / min until the rice straw bowl tray broke, as shown in Fig 5. The rice straw bowl tray deformation and stress change curves are automatically recorded and saved by the computer, as shown in Fig 6. According to the test stress-displacement curve, the pressure rod has no contact with the rice straw bowl tray at the initial state, and the contact stress is zero, after the pressure on the rice straw bowl tray, the test pressure gradually increased with the displacement. When the displacement reaches 23.3mm, the test stress exceeds the allowable stress of the rice straw bowl tray and breaks, and the test stress gradually decreased after the rice straw bowl tray is broken, the pressure rod is no longer stressed and the test stress is reduced to zero. The fracture point of the rice straw bowl tray corresponding to the curve peak during the test, and the corresponding fracture stress can indicate the rupture-resisting strength of the rice straw bowl tray.

**2.3.4 Test design.** Combined with the above analysis results, considering the agricultural technical requirements of rice straw bowl tray production, and test with vacuum degree ($X_1$), pressure preservation time ($X_2$) and adsorption time ($X_3$) as the influencing factors, test factor-level encoding table is shown in Table 2. Triests were performed according to the orthogonal rotation combination test scheme of ternary secondary regression. The vacuum degree of the pneumatic forming machine can be adjusted through the valve, the data is read and recorded by Asmik digital pressure gauge, the pressure protection time and adsorption time can be changed through the parameter setting interface of the control panel, and processing was averaged after 5 replicates for each group. The trial results were used for regression

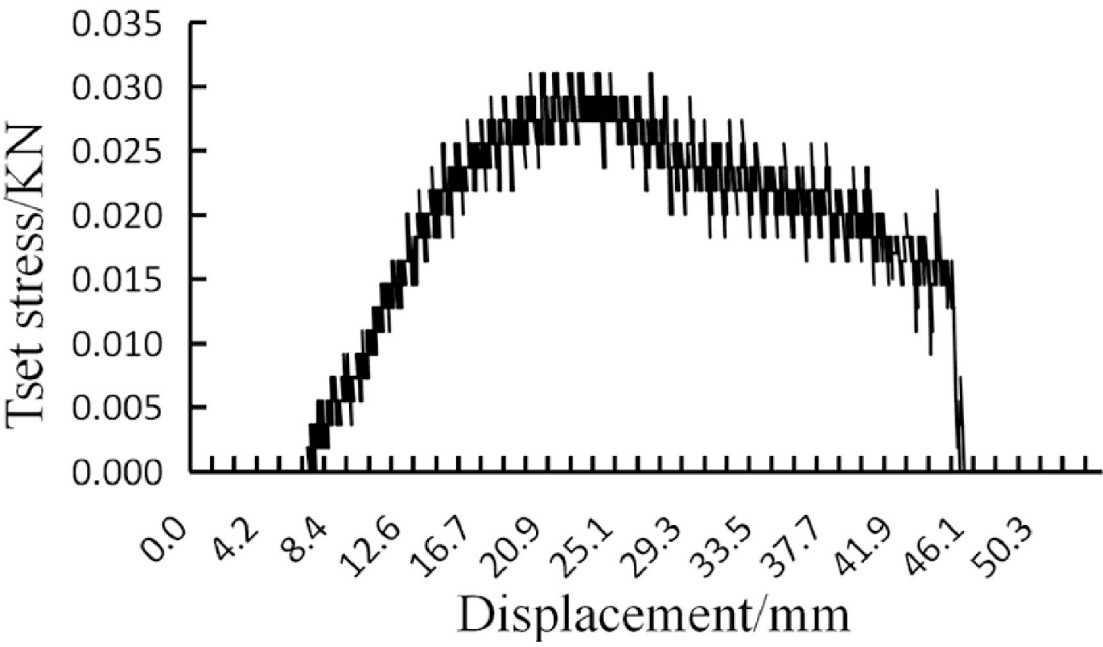

**Fig 6. Stress-displacement curve.**

analysis by Design Expert data analysis software to establish regression equations and were fit and significance tested, established the regression model of rice straw bowl tray molding performance, analyzed the influence of various factors and interactions on molding performance based on response surface analysis, and finally determined the best process parameters through the optimization module.

## 3. Results and optimization

### 3.1 Test results

In 2020, the trial was implemented at the Circular Agriculture Research Center. The test was conducted according to the central composite test design theory, and the test protocols and results are shown in Table 3.

For the obtained sample data, multiple regression fitting analysis was performed by Design-Expert data analysis software to establish a quadratic polynomial regression model of bowl hole molding rate $Y_1$, relaxation density $Y_2$ and rupture-resisting strength $Y_3$ for independent variables, as shown in Eq 3. The F test and variance analysis were performed on the established

**Table 2. Factor-level encoding table.**

| Levels | Code value | Natural variable | | |
|---|---|---|---|---|
| | | Vacuum degree $X_1$ | Pressure preservation time $X_2$ | Adsorption time $X_3$ |
| Top asterisk arm | +1.682 | -0.1 | 20 | 6 |
| Top level | +1 | -0.09 | 18 | 5 |
| Zero level | 0 | -0.075 | 15 | 4 |
| Low level | -1 | -0.04 | 12 | 3 |
| Low asterisk arm | -1.682 | -0.05 | 10 | 2 |

**Table 3. Test protocols and results.**

| Number | Experimental factors | | | Experiment indicators | | |
|---|---|---|---|---|---|---|
| | Vacuum degree $X_1$ | Pressure preservation time $X_2$ | Adsorption time $X_3$ | Bowl hole molding rate $Y_1$/% | Relaxation density $Y_2$/ g•cm³ | Rupture-resisting strength $Y_3$/N |
| 1 | 1 | 1 | 1 | 90.89 | 5.42 | 25.81 |
| 2 | 1 | 1 | -1 | 91.18 | 5.67 | 26.90 |
| 3 | 1 | -1 | 1 | 94.51 | 5.73 | 27.41 |
| 4 | 1 | -1 | -1 | 96.01 | 5.99 | 28.32 |
| 5 | -1 | 1 | 1 | 91.59 | 5.43 | 26.02 |
| 6 | -1 | 1 | -1 | 96.84 | 6.02 | 28.57 |
| 7 | -1 | -1 | 1 | 90.53 | 5.60 | 26.71 |
| 8 | -1 | -1 | -1 | 98.66 | 6.22 | 28.96 |
| 9 | 1.682 | 0 | 0 | 87.31 | 5.23 | 24.76 |
| 10 | -1.682 | 0 | 0 | 97.73 | 6.08 | 28.86 |
| 11 | 0 | 1.682 | 0 | 94.14 | 5.62 | 26.77 |
| 12 | 0 | -1.682 | 0 | 97.61 | 6.17 | 28.79 |
| 13 | 0 | 0 | 1.682 | 93.27 | 5.82 | 27.51 |
| 14 | 0 | 0 | -1.682 | 95.73 | 5.96 | 28.24 |
| 15 | 0 | 0 | 0 | 97.92 | 5.65 | 26.88 |
| 16 | 0 | 0 | 0 | 97.91 | 5.67 | 26.88 |
| 17 | 0 | 0 | 0 | 95.39 | 5.52 | 26.14 |
| 18 | 0 | 0 | 0 | 97.99 | 5.66 | 26.91 |
| 19 | 0 | 0 | 0 | 97.98 | 5.68 | 26.90 |
| 20 | 0 | 0 | 0 | 97.91 | 5.67 | 26.88 |
| 21 | 0 | 0 | 0 | 99.03 | 5.53 | 26.21 |
| 22 | 0 | 0 | 0 | 97.82 | 5.67 | 26.86 |
| 23 | 0 | 0 | 0 | 96.84 | 5.62 | 26.57 |

regression model, and the analysis results are shown in Table 4.

$$\begin{cases} Y_1 = 97.65 + 2.39X_1 + 1.10X_2 + 0.67X_3 + 0.51X_1X_2 + 1.45X_1X_3 - 0.96X_2X_3 - 1.88X_1^2 - 0.69X_2^2 - 1.18X_3^2 \\ Y_2 = 5.63 + 0.23X_1 + 0.14X_2 + 0.051X_3 + 0.005X_1X_2 + 0.087X_1X_3 - 0.033X_2X_3 - 0.005X_1^2 + 0.08X_2^2 + 0.079X_3^2 \quad (3) \\ Y_3 = 26.70 + X_1 + 0.55X_2 + 0.22X_3 - 0.06X_1X_2 + 0.35X_1X_3 - 0.24X_2X_3 + 0.003X_1^2 + 0.34X_2^2 + 0.38X_3^2 \end{cases}$$

Where $X_1$ is vacuum degree, MPa; $X_2$ is pressure preservation time, s; $X_3$ is adsorption time, s, and similarly hereinafter.

As can be seen in Table 4, regression equation models of bowl hole molding rate $Y_1$, relaxation density $Y_2$ and rupture-resisting strength $Y_3$ meet P<0.01, indicating that the model regression is extremely significant; If aberrant term of $Y_1$, $Y_2$ and $Y_3$ meet P>0.05, indicating that the insufficient fit was not significant, and the regression equations have favorable fitting effect and practical analytical significance; The multiple correlation indices $R^2$ of $Y_1$, $Y_2$ and $Y_3$ are 0.94, 0.96 and 0.94 respectively, which are all greater than 0.93, indicating that the models can explain more than 93% of response value changes, the predicted value is highly correlated to actual value and test error is small. Therefore the model can be used to predict and analyze the test results.

Based on regression analysis, the order of influences factors on bowl hole molding rate $Y_1$ is: vacuum degree $X_1$ > pressure preservation time $X_2$ > adsorption time $X_3$, in which $X_1$、 $X_2$、$X_1X_3$、$X_1{}^2$ and $X_3{}^2$ have a extremely significant influences on $Y_1$, $X_3$、$X_2X_3$, and $X_2{}^2$

**Table 4. Variance analysis of response surface model.**

| Variation source | $Y_1$ | | | | $Y_2$ | | | | $Y_3$ | | | |
|---|---|---|---|---|---|---|---|---|---|---|---|---|
| | Sum of squares | Freedom | F | P value | Sum of squares | Freedom | F | P value | Sum of squares | Freedom | F | P value |
| Model | 211.98 | 9 | 22.40 | 0.0001 | 1.30 | 9 | 34.47 | 0.0001 | 24.11 | 9 | 25.19 | 0.0001 |
| $X_1$ | 78.27 | 1 | 74.43 | 0.0001** | 0.73 | 1 | 172.9 | 0.0001** | 13.73 | 1 | 129.1 | 0.0001** |
| $X_2$ | 16.58 | 1 | 15.76 | 0.0016** | 0.27 | 1 | 64.62 | 0.0001** | 4.12 | 1 | 38.70 | 0.0001** |
| $X_3$ | 6.15 | 1 | 5.85 | 0.0310* | 0.035 | 1 | 8.43 | 0.0123* | 0.68 | 1 | 6.40 | 0.0252* |
| $X_1X_2$ | 2.09 | 1 | 1.99 | 0.1820 | 0.0002 | 1 | 0.048 | 0.8306 | 0.029 | 1 | 0.27 | 0.6115 |
| $X_1X_3$ | 16.79 | 1 | 15.97 | 0.0015** | 0.061 | 1 | 14.59 | 0.0021** | 0.98 | 1 | 9.21 | 0.0096** |
| $X_2X_3$ | 7.39 | 1 | 7.03 | 0.0200 | 0.0085 | 1 | 2.01 | 0.1796 | 0.47 | 1 | 4.42 | 0.0555 |
| $X_1{}^2$ | 56.01 | 1 | 53.27 | 0.0001** | 0.0003 | 1 | 0.078 | 0.7851 | 0.001 | 1 | 0.001 | 0.9966 |
| $X_2{}^2$ | 7.60 | 1 | 7.22 | 0.0186 | 0.10 | 1 | 24.42 | 0.0003** | 1.87 | 1 | 17.61 | 0.0010** |
| $X_3{}^2$ | 22.03 | 1 | 20.95 | 0.0005** | 0.098 | 1 | 23.35 | 0.0003** | 2.26 | 1 | 21.22 | 0.0005** |
| Residual | 13.67 | 13 | | | 0.055 | 13 | | | 1.38 | 13 | | |
| Lack of fit | 5.54 | 5 | 1.09 | 0.4337 | 0.024 | 5 | 1.24 | 0.3754 | 0.61 | 5 | 1.25 | 0.3712 |
| Error | 8.13 | 8 | | | 0.031 | 8 | | | 0.78 | 8 | | |
| Correct total | 225.65 | 22 | | | 1.36 | 22 | | | 25.49 | 22 | | |

Note: $P<0.01$ (Extremely significant level**), $P<0.05$ (Significant*).

have a significant influences on $Y_1$, while other factors have insignificant influences on $Y_1$; The order of influences factors on relaxation density $Y_2$ is: vacuum degree $X_1$ = pressure preservation time $X_2$ > adsorption time $X_3$, in which $X_1$, $X_2$, $X_1X_3$, $X_2{}^2$ and $X_3{}^2$ have a extremely significant influences on $Y_2$, $X_3$ has a significant influence on $Y_2$ and other factors have insignificant influences on $Y_2$; The order of influences factors on rupture-resisting strength $Y_3$ is: vacuum degree $X_1$ = pressure preservation time $X_2$ > adsorption time $X_3$, in which $X_1$, $X_2$, $X_1X_3$, $X_2{}^2$ and $X_3{}^2$ have a extremely significant influences on $Y_3$, $X_3$ has a significant influence on $Y_3$, and other factors have insignificant influences on $Y_3$; Excluded the insignificant factors satisfying $P>0.05$, the model is refitted to obtain the optimized regression equation, as shown in Eq (4).

$$\begin{cases} Y_1 = 60.92 + 430.47X_1 + 1.43X_2 + 1.96X_3 + 28.98X_1X_3 - 0.1X_2X_3 - 3004.09X_1^2 - 0.03X_2^2 - 0.29X_3^2 \\ Y_2 = 5.97 + 2.22X_1 - 0.07X_2 - 0.26X_3 + 1.75X_1X_3 - 3.21X_2^2 - 0.02X_3^2 \\ Y_3 = 28.29 + 12.11X_1 - 0.30X_2 - 1.17X_3 + 7X_1X_3 + 0.01X_2^2 + 0.09X_3^2 \end{cases} \quad (4)$$

## 3.2 Response surface analysis

According to formula (4), $X_1X_3$ and $X_2X_3$ have a significant impact on $Y_1$, $X_1X_3$ has a significant impact on $Y_2$, and $X_1X_3$ has a significant impact on $Y_3$. In order to clarify the impact of interaction on each index, response surface analysis was carried out by using Design-Expert data analysis software.

When pressure preservation time is 15 s, interactive influence of vacuum degree and adsorption time on bowl hole molding rate is shown in Fig 7. It can be known that when the vacuum degree is less than -0.07MPa, the bowl hole molding rate varies relatively smoothly with the change of adsorption time, when the vacuum degree is greater than -0.07MPa, the bowl hole molding rate increased dramatically with the increasing of adsorption time. This is because in the low vacuum degree interval, the vacuum suction is also small, and the material

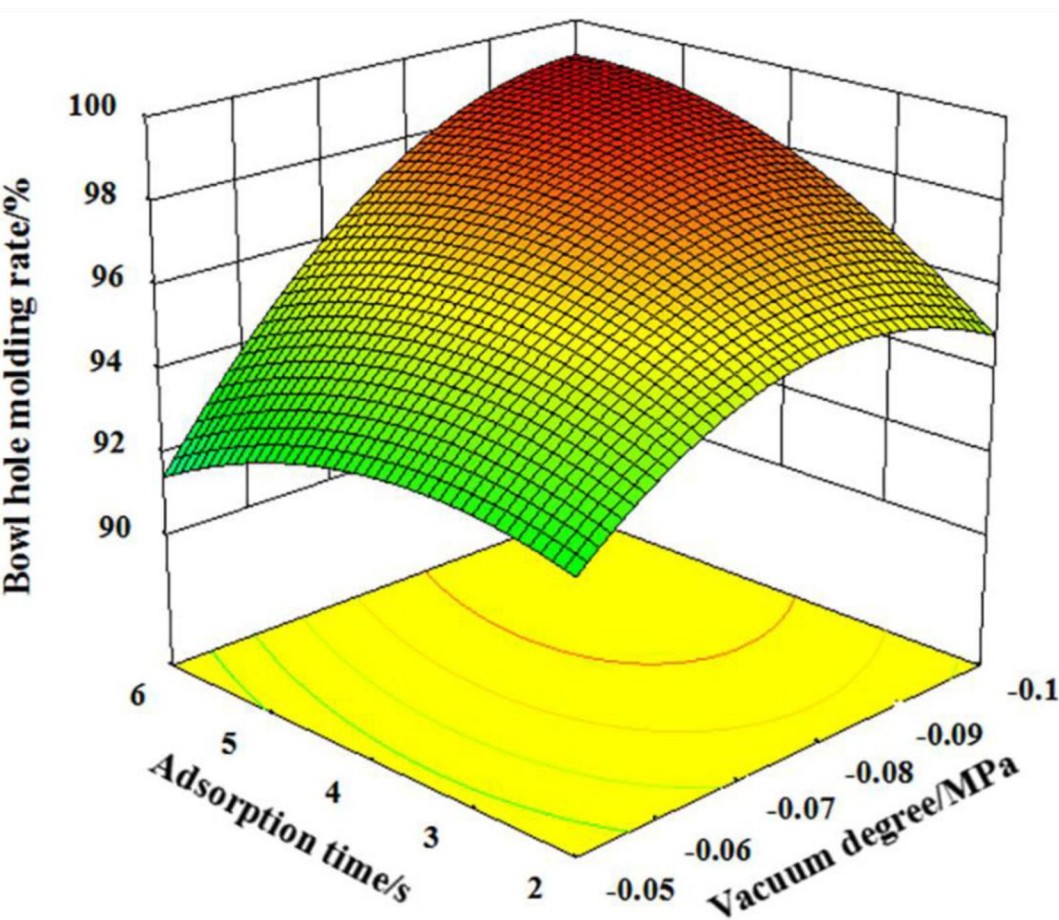

**Fig 7. The interaction of vacuum degree and adsorption time on the bowl hole molding rate.**

quality absorbed on the template is correspondingly less. At this time, even increasing the adsorption time, it is difficult to improve the quality of the adsorption materials, the thickness of the side wall of some bowls can not meet the design requirements and it is easy to damage when taking the mold. In the high vacuum degree interval, the adsorption force is enhanced, with the increasing of the adsorption time, the quality of adsorption material also increased, and the bowl hole molding rate is also improved. When the adsorption time is in the 2 to 3.5s interval, the bowl hole rate increased first and then decreased with the increases of vacuum degree, when the adsorption time is in the 3.5 to 6s interval, the bowl hole molding rate increases gradually with the increases of vacuum degree. This is because when the adsorption time is less than 3.5s, the adsorption amount on the template surface is less due to the insufficient adsorption time, the material and the mold are not closely combined, it is easy to damage during demoulding and if the vacuum degree is large, the limited material and the template combination is too tight, it is difficult to demolding. Therefore, a higher bowl hole molding rate can only be achieved only in appropriate scope. When the adsorption time is greater than 3.5s, the mold adsorption material time is sufficient, and the bowl hole molding rate gradually improved with the increasing of the vacuum degree.

When vacuum degree is -0.075 MPa, interactive influence of pressure preservation time and adsorption time on bowl hole molding rate is shown in Fig 8. It can be known that bowl

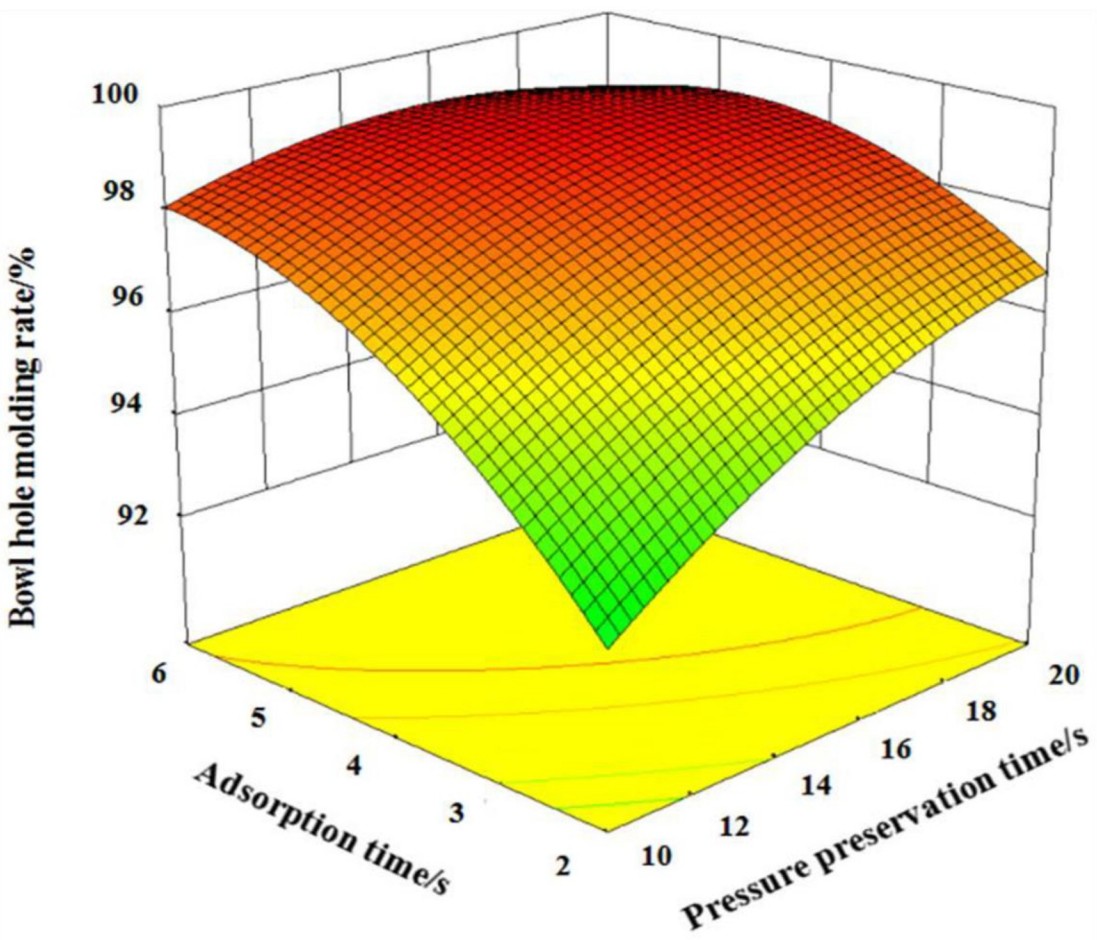

**Fig 8. The interaction of the pressure preservation time and adsorption time on the bowl hole molding rate.**

hole molding rate gradually increased with increasing of pressure preservation time and the adsorption time. Because the longer the adsorption time, the more materials on the surface of template, the greater the connection force between the materials, and the easier it is to form rice straw bowl tray. The longer the pressure preservation time is, the longer the solidification time of the material is, and the denser the material is, which is conductive to meet the strength requirements of the rice straw bowl tray, and then improving the bowl hole molding rate.

When pressure preservation time is 15 s, interactive influence of vacuum degree and adsorption time on relaxation density is shown in Fig 9. It can be known that when the vacuum degree is less than -0.085 MPa, the relaxation density gradually increased with the increasing of vacuum degree and adsorption time, and the relaxation density does not change much with the adsorption time. When the vacuum degree is greater than -0.085MPa, the relaxation density increased rapidly with the increasing of adsorption time. When the vacuum density increased, the relaxation density always maintains a high growth rate. Combining with the variance analysis results showed that this is because the vacuum degree affects the relaxation density significantly than the adsorption time. The adsorption time cannot effectively improve the relaxation density in the range of low vacuum degrees, when the appropriate vacuum degree range is reached, with the increase of vacuum degree and the adsorption, the template surface materials are closely combined and the relaxation density increased.

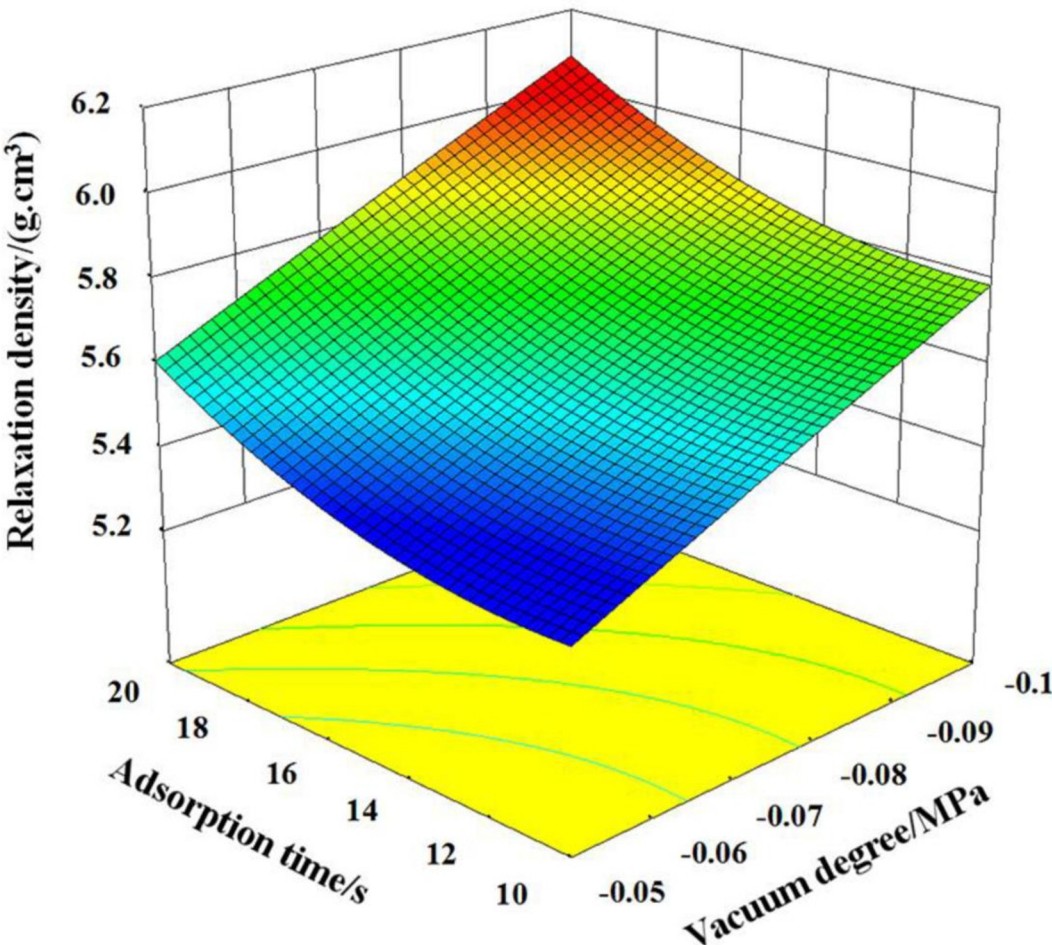

**Fig 9. The interaction of the adsorption time and vacuum degree on the relaxation density.**

When pressure preservation time is 15 s, interactive influence of vacuum degree and adsorption time on rupture-resisting strength is shown in Fig 10. It can be known that influencing trend of vacuum degree and adsorption time on the rupture-resisting strength is basically the same as that on the relaxation density, this is because there is a strong correlation between relaxation density and rupture-resisting strength, the greater the relaxation density of rice straw bowl tray, the higher the rupture-resisting strength, on the contrary, the smaller the relaxation density of rice straw bowl tray, the lower the rupture-resisting strength. Therefore, the change law of rupture-resisting strength can be used to explain the change of relaxation density.

### 3.3 Parameter optimization and verification test

In order to obtain optimal molding technological parameters of rice straw bowl tray, the optimization module of the design expert software is used to solve the constrained objective optimization of the regression model in this paper, because the higher the bowl hole molding rate $Y_1$, the higher the production efficiency, and the greater the rupture-resisting strength $Y_3$, the lower the damage rate in transport, therefore, $Y_1$ and $Y_3$ are required to take the maximum value. Early analysis showed that when relaxation density of rice straw bowl tray was 4 to 5.8 g/

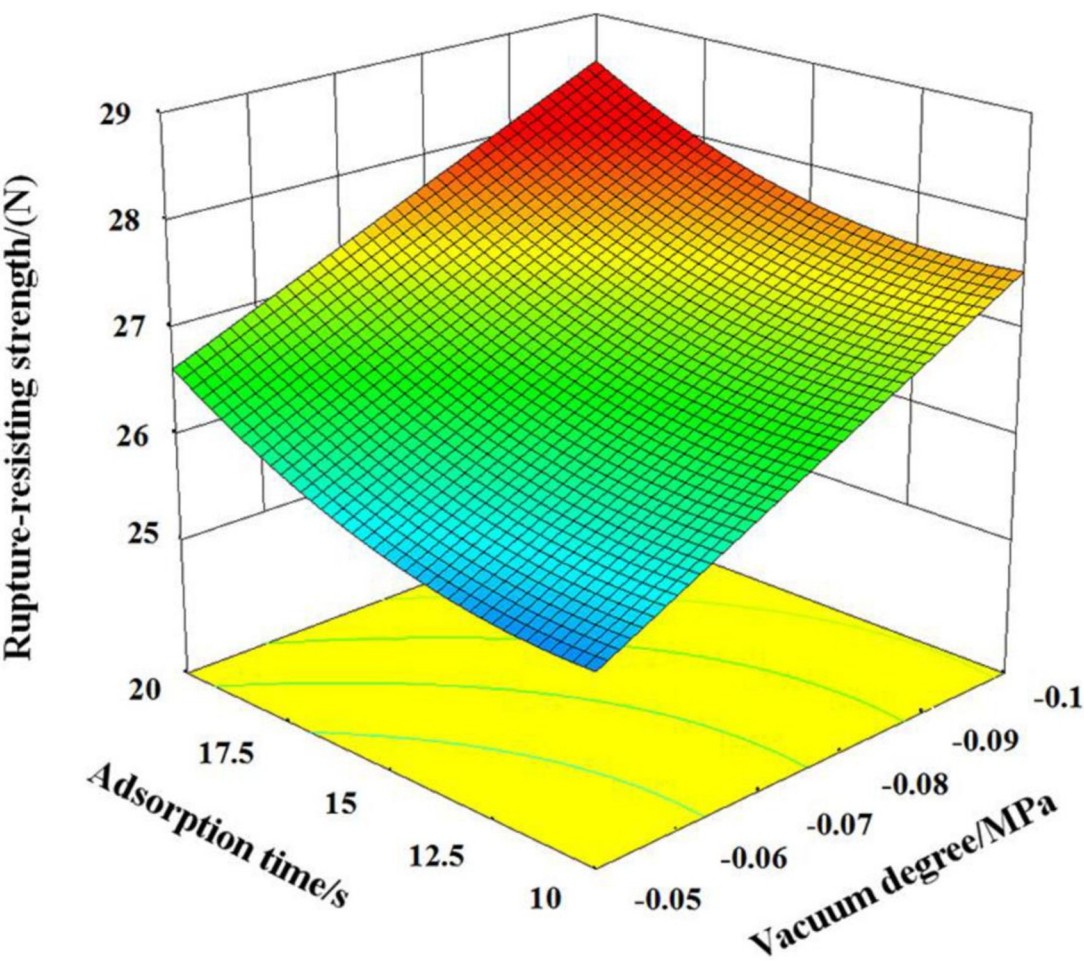

**Fig 10. The interaction of the adsorption time and vacuum degree on the rupture-resisting strength.**

cm$^3$, it was conductive to the growth of seeding roots and the bowl tray was not easy to disperse, $Y_2$ is therefore required to be in the range of 4 to 5.8 g/cm$^3$. The final determination objective function is shown in Eq (5).

$$\text{Objective function}: F = \begin{cases} \max Y_1(x_1, x_2, x_3) \\ 4 \leq Y_2(x_1, x_2, x_3) \leq 5.8 \\ \max Y_3(x_1, x_2, x_3) \end{cases} \quad (5)$$

Constraint function: 0.75<$X_1$<0.1; 10<$X_2$<20; 3.5<$X_3$<6.

Through the analysis, the optimization results are as follows: the vacuum degree is -0.09 MPa, the pressure preservation time is 14 s, and the adsorption time is 5 s. in this state, the theoretical bowl hole molding rate is 98.62%, relaxation density is 5.8 g/cm$^3$ and rupture-resisting strength is 27.48 N, meet the design and agricultural technology requirements of rice straw bowl tray.

In order to verify the reliability of optimization and actual production results, optimal technological parameters obtained through model optimization were used for verification test, a total of 20 repeated tests were totally set, and average values of the measured parameter values were recorded in Table 5. It could be known from the table that relative errors between

**Table 5. Experiment data for test validation.**

| Parameter | bowl hole molding rate/% | Relaxation density/(g·cm$^{-3}$) | Rupture-resisting strength/N |
|---|---|---|---|
| Predicted value | 98.62 | 5.8 | 27.48 |
| Mean of test | 99.28 | 5.4 | 26.94 |
| Relative error/% | 6 | 7.5 | 2 |

Note: Vacuum degree = -0.09 MPa, pressure preservation time = 14 s, adsorption time = 5 s.

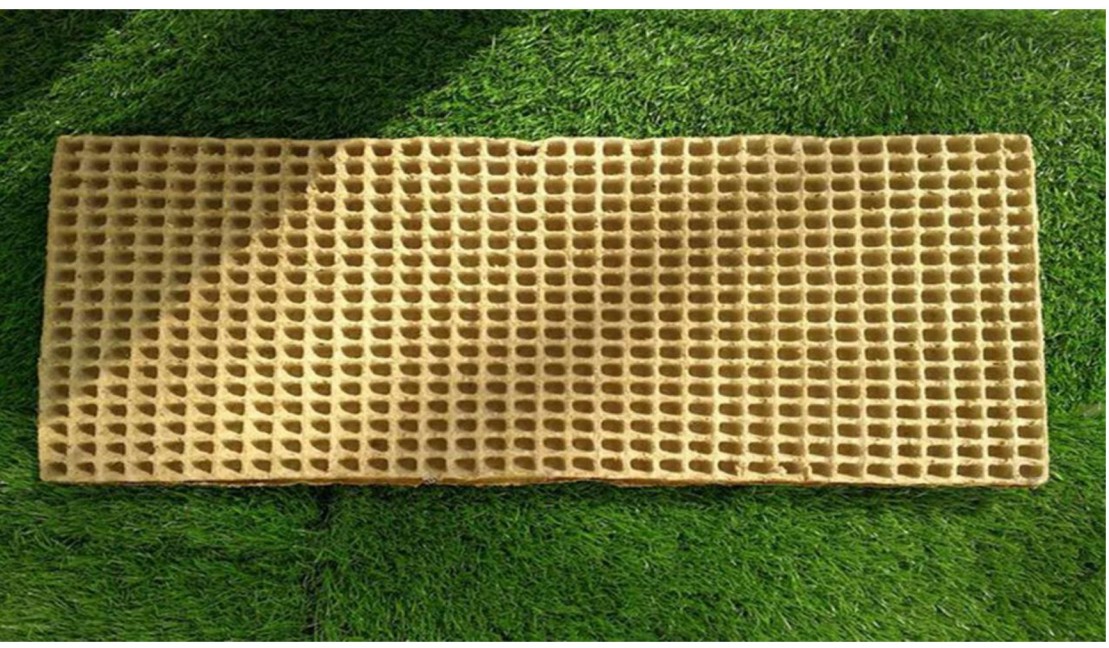

**Fig 11. Picture of rice straw bowl tray.**

measured values and predicted values of bowl hole molding rate, relaxation density and rupture-resisting strength were 6%, 7.5% and 2% respectively, so relative errors between measured and predicted results were small with favorable model fitting degree, and this certified that regression equations of rice straw bowl tray molding technology established in this study were reliable, and regression equations could be used to accurately predict the test results. Meanwhile, it could be known by observing appearance quality of rice straw bowl tray that it was of uniform color and complete molding, without deformation or crack. Due to the existence of the filified straw in the raw material, cause the edges of the rice straw bowl tray were hairy, however, the overall strength and use performance of the rice straw bowl tray were not affected. The rice straw bowl tray was shown in Fig 11.

## 4. Conclusions

1. This study carried out multivariate analysis of production technology of rice straw bowl tray, and studied influences of vacuum degree, adsorption time and pressure preservation time on bowl hole molding rate, relaxation density and rupture-resisting strength. The order of influences factors on bowl hole molding rate is: vacuum degree > pressure

preservation time > adsorption time; the order of influences factors on the relaxation density and rupture-resisting strength is: vacuum degree = pressure preservation time > adsorption time; and the regression model between molding technology and molding performance was established using Design-Expert data analysis software.

2. When the vacuum degree, pressure preservation time and adsorption time were -0.09 MPa, 14 s and 5 s respectively, the molding effect of rice straw bowl tray was optimal. At this time, bowl hole molding rate, relaxation density and rupture-resisting strength were 98.62%, 5.8 g/cm$^3$ and 27.48 N respectively. Through the verification test, the relative error between the measured value and the predicted value was less than 8%, and the obtained model had high reliability.

3. The results of this study are of great significance to improve the production quality of rice straw bowl tray, improve the high efficiency of rice straw bowl tray and reduce the production cost. Moreover, it provides theoretical basis for the factory production of the rice straw bowl tray and a new direction for the efficient utilization of rice straw.

## Supporting information

**S1 Data. Test protocols and results.**
(XLSX)

## Author Contributions

**Conceptualization:** Xinyu Liu.

**Data curation:** Huafen Zou, Huiyuan Wang, Zhenzhen Yu.

**Investigation:** Haitian Sun.

**Methodology:** Hongxuan Wang.

**Project administration:** Hailiang Li, Chun Wang, Xinyu Liu.

**Software:** Hongxuan Wang.

**Supervision:** Xinyu Liu.

**Validation:** Chun Wang.

**Writing – original draft:** Hailiang Li, Huiyuan Wang, Xinyu Liu.

**Writing – review & editing:** Huiyuan Wang, Xinyu Liu, Zhenzhen Yu.

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
