## [Decision Letter · Decision Letter 0]

3 Nov 2021

PONE-D-21-20345Optimization of Technique Parameters of Pneumatic Molding Machine for Nutritive Rice Straw Potted TrayPLOS ONE

Dear Dr. YU,

Thank you for submitting your manuscript to PLOS ONE. After careful consideration, we feel that it has merit but does not fully meet PLOS ONE’s publication criteria as it currently stands. Therefore, we invite you to submit a revised version of the manuscript that addresses the points raised during the review process.

We look forward to receiving your revised manuscript.

Kind regards,

Gabriele Oliva, Ph.D

Academic Editor

PLOS ONE

Journal Requirements:

2. PLOS requires an ORCID iD for the corresponding author in Editorial Manager on papers submitted after December 6th, 2016. Please ensure that you have an ORCID iD and that it is validated in Editorial Manager. To do this, go to ‘Update my Information’ (in the upper left-hand corner of the main menu), and click on the Fetch/Validate link next to the ORCID field. This will take you to the ORCID site and allow you to create a new iD or authenticate a pre-existing iD in Editorial Manager. Please see the following video for instructions on linking an ORCID iD to your Editorial Manager account: https://www.youtube.com/watch?v=_xcclfuvtxQ.

4. Please include captions for your Supporting Information files at the end of your manuscript, and update any in-text citations to match accordingly. Please see our Supporting Information 

Additional Editor Comments (if provided):

Two reviews were collected, both suggesting major revision. After reviewing the paper myself, I agree with the reviewers' evaluation.

Reviewers' comments:

Reviewer's Responses to Questions

**Comments to the Author**

1. Is the manuscript technically sound, and do the data support the conclusions?

Reviewer #1: Yes

Reviewer #2: Partly

2. Has the statistical analysis been performed appropriately and rigorously? 

Reviewer #1: Yes

Reviewer #2: Yes

3. Have the authors made all data underlying the findings in their manuscript fully available?

Reviewer #1: Yes

Reviewer #2: No

4. Is the manuscript presented in an intelligible fashion and written in standard English?

Reviewer #1: No

Reviewer #2: No

5. Review Comments to the Author

Reviewer #1: The article provides a theoretical basis for optimization of pneumatic molding technological parameters of nutritive rice straw potted tray. Even if the contribution is clear and the article is well structured, there are some issues which have to be solved:

1) In general, the English is too poor: the entire article needs improvement in writing

2) Abstract has to be more general without reporting technical data which will be addressed in the following sections. Thus, abstract rewriting is necessary

3) The state of the art has to be improved by providing a higher number of references.

Based on the comments above, I would like to accept this paper if my concerns are carefully addressed.

Reviewer #2: The paper entitled “Optimization of Technique Parameters of Pneumatic Molding Machine for Nutritive Rice Straw Potted Tray” is a study that proposes a multi factor analysis of production technology of nutritive rice straw potted tray. The study aims at investigating the influence of 3 molding parameters (i.e., vacuum degree, holding time of molding die and absorption time) on 3 molding indices (i.e., pot hole molding rate, relaxation density and rupture-resisting strength). This paper findings could be helpful to optimize the pneumatic molding process in view of an industrialized production of nutritive rice straw potted tray, however there are some important aspects that need to be addressed before publication:

The authors need to revise the entire manuscript carefully and correct the multiple grammar and spelling mistakes. The overall English quality and the multiple typos make the reading of the paper really hard. For instance:

- “demolding damage root systems” (pag. 2 line 4)

- “Maximum stress borne” (pag. 9 line 14)

- “research institute chinese academy of tropical agriculture sciences” (page 3)

- Check Table 3 Header

There is a lack of a proper state of the art section. In the introduction section you cite Teng Cuiquing’s study but it is not clear which results were obtained in the degradation property test. Moreover, you mention a few other studies without underlying which are the advantages and limitations of each type of potted tray.

In the Introduction Section it is not clear in which way pneumatic molding mode may solve the problems you stated above.

In the Introduction section you do not say in which proportion straws were blended with raw materials. It is not clear the experimental setup and how long the seriflux was preserved in the seriflux pond.

At Pag. 5 - you make a list of the components of the machine. Please describe a general process of pneumatic molding and how each system is used in the process.

In the Test method Section you should specify in which proportion water and adhesive were placed in seriflux and you should consider adding a few lines to explain the testing procedure.

6. PLOS authors have the option to publish the peer review history of their article (what does this mean?). If published, this will include your full peer review and any attached files.

Reviewer #1: No

Reviewer #2: No

---

## [Author Response · Author response to Decision Letter 0]

28 Apr 2022

Reviewer #1: The article provides a theoretical basis for optimization of pneumatic molding technological parameters of nutritive rice straw potted tray. Even if the contribution is clear and the article is well structured, there are some issues which have to be solved:

Reviewers' comments:

1)In general, the English is too poor: the entire article needs improvement in writing

2)Abstract has to be more general without reporting technical data which will be addressed in the following sections. Thus, abstract rewriting is necessary

3)The state of the art has to be improved by providing a higher number of references.

Reviewer's Responses to Questions

1)The entire paper has been revised and professionally polished in English.

2)The paper abstract is simplified and the technical data content is removed.

3)The number of relevant references was increased.

Reviewer #2: The paper entitled “Optimization of Technique Parameters of Pneumatic Molding Machine for Nutritive Rice Straw Potted Tray” is a study that proposes a multi factor analysis of production technology of nutritive rice straw potted tray. The study aims at investigating the influence of 3 molding parameters (i.e., vacuum degree, holding time of molding die and absorption time) on 3 molding indices (i.e., pot hole molding rate, relaxation density and rupture-resisting strength). This paper findings could be helpful to optimize the pneumatic molding process in view of an industrialized production of nutritive rice straw potted tray, however there are some important aspects that need to be addressed before publication:

Reviewers' comments:

1)The authors need to revise the entire manuscript carefully and correct the multiple grammar and spelling mistakes. The overall English quality and the multiple typos make the reading of the paper really hard. 

2)There is a lack of a proper state of the art section. In the introduction section you cite Teng Cuiquing’s study but it is not clear which results were obtained in the degradation property test. Moreover, you mention a few other studies without underlying which are the advantages and limitations of each type of potted tray.

3)In the Introduction Section it is not clear in which way pneumatic molding mode may solve the problems you stated above.

4)In the Introduction section you do not say in which proportion straws were blended with raw materials. It is not clear the experimental setup and how long the seriflux was preserved in the seriflux pond.

5)At Pag. 5 - you make a list of the components of the machine. Please describe a general process of pneumatic molding and how each system is used in the process.

6)In the Test method Section you should specify in which proportion water and adhesive were placed in seriflux and you should consider adding a few lines to explain the testing procedure.

Reviewer's Responses to Questions

1)The entire paper has been revised and professionally polished in English.

2)In the second paragraph of the introduction, it was discussed on biomass seedling bowl, which shows the characteristics and problems of different kinds of biomass seedling bowl, which provides the foundation for nutritive rice straw bowl.

3)The third paragraph of the introduction illustrated the problems of the pressing forming method, and presented the principle and advantages of the pneumatic forming method.

4)The material pretreatment method and mixing ratio used in the test are reflected in the Test materials part, which is not reflected in the introduction to avoid repetition. The slurry can be stored in the slurry tank and can be used to prepare nutritive rice straw bowl at any time after uniform mixing. 

5)The general process of pneumatic molding and how each system is used in the process were described in the part of Production method of potted tray. 

6)The proportion of water and adhesive during the slurry preparation is expressed in lines 105-107 in the article

---

## [Decision Letter · Decision Letter 1]

24 May 2022

PONE-D-21-20345R1Optimization of Technique Parameters of Pneumatic Molding Machine for Nutritive  Rice Straw BowlPLOS ONE

Dear Dr. YU,

Thank you for submitting your manuscript to PLOS ONE. After careful consideration, we feel that it has merit but does not fully meet PLOS ONE’s publication criteria as it currently stands. Therefore, we invite you to submit a revised version of the manuscript that addresses the points raised during the review process.

We look forward to receiving your revised manuscript.

Kind regards,

Gabriele Oliva, Ph.D

Academic Editor

PLOS ONE

Additional Editor Comments (if provided):

One reviewer is still not convinced and raises some major comments. I invite the authors to prepare a major revision to address such comments (or to provide a compelling discussion why some of them were not addressed).

Notice that it is my intention to take a final decision based on the next round of revision.

Reviewers' comments:

Reviewer's Responses to Questions

**Comments to the Author**

1. If the authors have adequately addressed your comments raised in a previous round of review and you feel that this manuscript is now acceptable for publication, you may indicate that here to bypass the “Comments to the Author” section, enter your conflict of interest statement in the “Confidential to Editor” section, and submit your "Accept" recommendation.

Reviewer #1: All comments have been addressed

Reviewer #2: (No Response)

2. Is the manuscript technically sound, and do the data support the conclusions?

Reviewer #1: Yes

Reviewer #2: Partly

3. Has the statistical analysis been performed appropriately and rigorously? 

Reviewer #1: Yes

Reviewer #2: Yes

4. Have the authors made all data underlying the findings in their manuscript fully available?

Reviewer #1: Yes

Reviewer #2: Yes

5. Is the manuscript presented in an intelligible fashion and written in standard English?

Reviewer #1: Yes

Reviewer #2: No

6. Review Comments to the Author

Reviewer #1: The article provides a theoretical basis for optimization of pneumatic molding technological parameters of nutritive rice straw potted tray. The paper is now well-revised (specifically in the English writing) and thus it can be accepted for PLOS ONE publication.

Reviewer #2: This paper proposes regression models between potted tray molding factors and molding properties to obtain the optimal technological parameters for the pneumatic molding process. The paper reports results that could be relevant in the optimization of pneumatic molding process for nutritive rice straws potted trays. The paper shows well-structured quantitative results and a well thought experimental analysis. However, the way the paper is written, in terms of structure of the sentences and poor English, often invalidates the fluency of the reading and the sense of the paper, preventing one from understanding its meaning and objective. In my opinion, the paper needs major revisions in terms of both written language and unclear points before publication on PLOSONE journal. Some of the unclear points or concepts that should be better explained are listed below:

- In the state-of-the-art section you mainly explain the raw material used by other research groups to prepare the seedling bowl, but you scarcely report quantitative results in the existing bowls performance with respect to the parameters you decided to take into consideration. Moreover, you should highlight if any other study considered the same or similar quantitative performance parameters.

- Explain what you mean by straw quality score.

- In the text should emerge before why you chose to evaluate the vacuum degree, pressure preservation time and adsorption time factors and why they are so relevant during the molding process. Try to briefly explain the concept when you first introduce the factors you decided to evaluate.

- It is not clear what you mean by bowl forming rate.

- In the response surface test paragraph, you should insert a few lines at the beginning of the paragraph to introduce the reader to what this part of the paper is about, what tests will be explained in the paragraph and what parameters change from one test to the other.

- In the response surface test paragraph, what is the software used to obtain the model shown in the figures?

- Define in the text the acronym ANOVA before using it.

A few expamples of unclear senstences/ misused English terms:

- Line 3: light quality.

- Line 3: The commonly used plastic plate mostly made of polyethylene, with low price, light quality and good water protection, but transplanting mold damage root system, not easy to degradation, improper recycling can cause environmental pollution.

- “..forming ways, not only improve the utilization rate of agricultural waste, give full play to the residual value of agricultural waste, but also solve the agricultural waste and environmental pollution.”

- Be sure each sentence makes sense. Make shorter sentences if needed to prevent the loss of the key concept you want to express by the sentence.

7. PLOS authors have the option to publish the peer review history of their article (what does this mean?). If published, this will include your full peer review and any attached files.

Reviewer #1: No

Reviewer #2: No

---

## [Author Response · Author response to Decision Letter 1]

29 May 2022

（1）In the state-of-the-art section you mainly explain the raw material used by other research groups to prepare the seedling bowl, but you scarcely report quantitative results in the existing bowls performance with respect to the parameters you decided to take into consideration. Moreover, you should highlight if any other study considered the same or similar quantitative performance parameters.

Response：In this paper, the research of biomass seedling bowl molding technology and molding process are discussed, and the research status of pneumatic forming technology is explained.

（2） Explain what you mean by straw quality score.

Response：The explanation of straw quality score appears in the first paragraph of “2.3.1 Production method of potted tray”.

（3） In the text should emerge before why you chose to evaluate the vacuum degree, pressure preservation time and adsorption time factors and why they are so relevant during the molding process. Try to briefly explain the concept when you first introduce the factors you decided to evaluate.

Response：The reasons for selecting vacuum degree, adsorption time and pressure preservation time as experimental factors and their important correlation to the forming effect of seedling tray are described in the first three sections of “2.3.2 test factors”.

（4） It is not clear what you mean by bowl forming rate.

Response：Bowl forming rate is the bowl hole molding rate, the mean of bowl hole molding rate and the calculation formula are detailed in the second paragraph of “2.3.3 Test indices and measurement method”.

（5） In the response surface test paragraph, you should insert a few lines at the beginning of the paragraph to introduce the reader to what this part of the paper is about, what tests will be explained in the paragraph and what parameters change from one test to the other.

Response：The main contents of response surface analysis have been introduced in the first paragraph of “3.2 Response surface analysis”.

（6） In the response surface test paragraph, what is the software used to obtain the model shown in the figures?

Response：The software used for response surface analysis has been described in the first paragraph of “3.2 Response surface analysis”.

（7） Define in the text the acronym ANOVA before using it.

Response：ANOVA has been replaced with variance analysis.

（8）A few expamples of unclear senstences/ misused English terms:

- Line 3: light quality.

- Line 3: The commonly used plastic plate mostly made of polyethylene, with low price, light quality and good water protection, but transplanting mold damage root system, not easy to degradation, improper recycling can cause environmental pollution.

- “..forming ways, not only improve the utilization rate of agricultural waste, give full play to the residual value of agricultural waste, but also solve the agricultural waste and environmental pollution.”

Response：The problems of unclear senstences/ misused English terms were revised.

（9） Be sure each sentence makes sense. Make shorter sentences if needed to prevent the loss of the key concept you want to express by the sentence.

Response：Inappropriate sentences in the text have been corrected.

---

## [Decision Letter · Decision Letter 2]

10 Jun 2022

Optimization of Technique Parameters of Pneumatic Molding for Rice Straw Bowl  Tray

PONE-D-21-20345R2

Dear Dr. YU,

We’re pleased to inform you that your manuscript has been judged scientifically suitable for publication and will be formally accepted for publication once it meets all outstanding technical requirements.

Kind regards,

Gabriele Oliva, Ph.D

Academic Editor

PLOS ONE

Additional Editor Comments (optional):

The two reviewers recommend acceptance. I agree with them.

Reviewers' comments:

Reviewer's Responses to Questions

**Comments to the Author**

1. If the authors have adequately addressed your comments raised in a previous round of review and you feel that this manuscript is now acceptable for publication, you may indicate that here to bypass the “Comments to the Author” section, enter your conflict of interest statement in the “Confidential to Editor” section, and submit your "Accept" recommendation.

Reviewer #1: All comments have been addressed

Reviewer #2: All comments have been addressed

2. Is the manuscript technically sound, and do the data support the conclusions?

Reviewer #1: Yes

Reviewer #2: Yes

3. Has the statistical analysis been performed appropriately and rigorously? 

Reviewer #1: Yes

Reviewer #2: Yes

4. Have the authors made all data underlying the findings in their manuscript fully available?

Reviewer #1: Yes

Reviewer #2: Yes

5. Is the manuscript presented in an intelligible fashion and written in standard English?

Reviewer #1: Yes

Reviewer #2: Yes

6. Review Comments to the Author

Reviewer #1: (No Response)

Reviewer #2: (No Response)

7. PLOS authors have the option to publish the peer review history of their article (what does this mean?). If published, this will include your full peer review and any attached files.

Reviewer #1: No

Reviewer #2: No

---

## [Editor Report · Acceptance letter]

26 Aug 2022

PONE-D-21-20345R2 

Optimization of Technique Parameters of Pneumatic Molding for Rice Straw Bowl Tray 

Dear Dr. Yu:

I'm pleased to inform you that your manuscript has been deemed suitable for publication in PLOS ONE. Congratulations! Your manuscript is now with our production department. 

Kind regards, 

on behalf of

Dr. Gabriele Oliva 

Academic Editor

PLOS ONE